# Clusters and associations of adverse neonatal events with adult risk of multimorbidity: A secondary analysis of birth cohort data

Jeeva John[1◉], Seb Stannard[1‡], Simon D. S. Fraser[1‡], Ann Berrington[2‡], Nisreen A. Alwan[1,3,4‡] *

1 School of Primary Care, Population Sciences and Medical Education, Faculty of Medicine, University of Southampton, Southampton, United Kingdom, 2 Department of Social Statistics and Demography, University of Southampton, Southampton, United Kingdom, 3 University Hospital Southampton National Health Service Foundation Trust, Southampton, United Kingdom, 4 National Institute for Health Research Applied Research Collaboration Wessex, Southampton, United Kingdom

◉ These authors contributed equally to this work.
‡ SS, SDSF, AB and NAA authors also contributed equally to this work.
* n.a.alwan@soton.ac.uk

## Abstract

### Objective

To investigate associations between clustered adverse neonatal events and later-life multimorbidity.

### Design

Secondary analysis of birth cohort data.

### Setting

Prospective birth cohort study of individuals born in Britain in one week of 1970.

### Population

Respondents provided data at birth (n = 17,196), age 34 (n = 11,261), age 38 (n = 9,665), age 42 (n = 9,840), and age 46 (n = 8,580).

### Methods

Mixed components analysis determined included factors, 'Birthweight'; 'Neonatal cyanosis'; 'Neonatal cerebral signs'; 'Neonatal illnesses'; 'Neonatal breathing difficulties'; and 'Prolonged duration to establishment of respiratory rate at birth', within the composite adverse neonatal event score. Log-binomial regression quantified the unadjusted and covariate-adjusted (paternal employment status and social class; maternal smoking status; maternal age; parity; cohort member smoking status and Body Mass Index) associations between the adverse neonatal event score and risk of multimorbidity in adulthood.

**Data availability statement:** The BCS70 datasets generated and analysed in the current study are available through the UK Data Archive repository (available here: http://www.cls.ioe.ac.uk/page.aspx?&sitesectionid=795).

**Funding:** This research forms part of the MELD-B project supported by the National Institute for Health Research (NIHR) Artificial Intelligence for Multiple and Long-Term Conditions (NIHR203988). The views expressed are those of the authors and not necessarily those of the NIHR or the Department of Health and Social Care. The funders had no role in study design, data collection and analysis, decision to publish, or preparation of the manuscript.

**Competing interests:** The authors have declared that no competing interests exist.

## Outcome measures

Multimorbidity at each adult data sweep, defined as the presence of two or more Long-Term Conditions (LTCs).

## Results

13.7% of respondents experienced one or more adverse neonatal event(s) at birth. The percentage reporting multimorbidity increased steadily from 14.6% at age 34 to 25.5% at age 46. A significant association was only observed at the 38 years sweep; those who had experienced two or more adverse neonatal events had a 41.0% (95% CI: 1.05 – 1.88) increased risk of multimorbidity, compared to those who had not suffered any adverse neonatal events at birth. This association was maintained following adjustment for parental confounders and adult smoking status.

## Conclusions

Adverse neonatal events at birth may be independently associated with the development of midlife multimorbidity. Programmes and policies aimed at tackling the growing public health burden of multimorbidity may also need to consider interventions to reduce adverse neonatal events at birth.

## Introduction

Multimorbidity is commonly defined as the co-occurrence of two or more Long-Term Conditions (LTCs) [1]. This phenomenon is growing in prevalence [2], including amongst younger age groups [3], and represents a significant public health issue [4]. An estimated one in three individuals will develop multimorbidity in their lifetimes [5]. Multimorbidity results in higher rates of premature mortality [6], reduced quality of life [7], and greater demand on health services [8]. Previous evidence has demonstrated that multimorbidity is a significant driver of costs in many health and social care systems, independent of biological factors such as advanced age [9]. Multimorbidity also potentiates health inequalities; it is well established that the prevalence of multimorbidity is higher, and the age of onset is up to ten years earlier, in the most disadvantaged communities [10], and amongst certain ethnic minorities [11]. It is therefore vital that risk factors of multimorbidity are identified to facilitate timely detection and management of susceptible individuals [12], and aid the development and implementation of preventative interventions [13].

Research has demonstrated that certain early-life characteristics are associated with multimorbidity in adulthood [14–22]. Amongst the 1970 British Cohort Study, variables in childhood including parental social class, birthweight, childhood BMI, cognitive ability and behaviour were associated to a count of multimorbidity at midlife [14]. For the Hertfordshire cohort study, higher rates of childhood illnesses were associated with future multimorbidity and higher medication counts at ages 64-68 [15]. Amongst a birth cohort born in Helsinki, individuals born to mothers under the age of 25, mothers with a BMI above 25, individuals who had a birthweight less than 2.5 kg, those with rapid growth in height and weight from birth until age 11, wartime separation and paternal occupational class were all associated with a faster rate of chronic disease accumulation from midlife onwards [16]. Other research has found that early childhood conditions including parental socioeconomic status [17–22], poor childhood health [17,21], child maltreatment [22], child adversity including abuse and

neglect [19], negative caregiver's characteristics [19], food restriction [21], child labour [21], and stressful life events [21], were associated with multimorbidity across the adult life course. However, with the exception of birthweight and maternal age, previous research has yet to explore the relationship between multiple adverse neonatal events and multimorbidity.

Previous studies, adopting a disease-specific model, have shown associations between certain neonatal adversities and single adult LTCs, including low birth weight and adult-onset cardiovascular disease, and preterm birth and mid-life diabetes [23,24]. This has led to the conclusion that health status in adulthood is influenced by a series of interconnected events, beginning in early-life [25,26]. For example, poor childhood health may affect educational attainment [27], which may precede lower socioeconomic status (SES) in adulthood [28], resulting in chronic stress [29], a known risk factor for LTCs [30].

The effect of early life events on multimorbidity can be broadly explained by two main life-course epidemiological paradigms: the "critical period" theory, in which biological imprinting at important time-points, make an individual more susceptible to compromised health in adult life [31] and; the "accumulation of risk" model, which states that cumulative adverse early-life events contribute to poor adult health [32]. Both theories have potential relevance to the aetiology of multimorbidity, and have therefore been considered as a potential mechanism for this study.

It is plausible that events surrounding the birth of an individual represent an opportunity for early intervention, with the aim to prevent later-life multimorbidity. Yet, the association between multiple adverse neonatal events such as gestational age, birthweight, neonatal resuscitation, neonatal cyanosis, neonatal cerebral signs, duration to establishment of respiratory rate at birth, neonatal cephalohematoma, neonatal illnesses and breathing difficulties, and adult risk of multimorbidity is under-researched. Additionally, despite advancements in neonatal care in recent decades [33], the sizeable global burden posed by poor birth outcomes has persisted [34]. Preterm birth rates have not seen a decline globally, constituting approximately 10% of all livebirths worldwide [35].

We hypothesised that an increased number of adverse neonatal events would be associated with a greater burden of multimorbidity across adulthood. By considering the outcome of multimorbidity at various ages between 34 and 46 years, we also investigated whether experiencing a greater number of adverse neonatal events at birth was associated with an earlier onset of multimorbidity.

## Aim & objectives

The aim of this study was to investigate the clusters, and later-life multimorbidity associations of adverse neonatal events at birth within the 1970 British Cohort Study (BCS70).

This aim was achieved via three main objectives:

1. Identify and characterise clustering of adverse neonatal events.

2. Develop an *adverse neonatal events* score for use as the main exposure.

3. Investigate the association between the *adverse neonatal events* score and risk of multimorbidity at ages 34, 38, 42 and 46, accounting for confounders and mediators.

## Methods

### Study design and population

The BCS70 is a prospective birth cohort study of 17,196 individuals born in Britain in one week of April 1970. Full cohort description can be found elsewhere [36]. This work uses data

from five sweeps: ages 0, 34, 38, 42 and 46, and a flow chart highlighting the sample size at each sweep is included in S1 Fig. All data were non-identifiable and accessed through the UK Data Service. Inclusion criteria were singleton births of all live infants recruited to the BCS70 study at the first data sweep. Births resulting in perinatal mortality, stillbirth, and early neonatal deaths were excluded from further analysis. Due to the association between congenital abnormalities and adult multimorbidity, infants with known congenital abnormalities (outlined in S1 Table) were excluded [37].

The study was conducted in accordance with the UK Policy Framework for Health and Social Care Research. Ethics approval has been obtained from the University of Southampton Faculty of Medicine Ethics committee (ERGO II Reference 66810).

## Exposure

Information on *adverse neonatal events* were collected for questionnaires that were completed by the midwives who had been present at the birth and, in addition, information was extracted from clinical records. Score was generated as a composite score of markers of poor outcomes at birth, described in previous literature a full break down of the scores is provided in (S2 Table) [38,39]. One point was scored every time an indicator listed below was present:

- Preterm birth (<259 days);

- Low Birthweight (<2500 grammes);

- Requirement for neonatal resuscitation;

- Presence of neonatal cyanosis;

- Presence of neonatal cerebral signs, including irritability, hypertonia, hypotonia, shrill cries, hypocalcaemia and other;

- Prolonged duration to establishment of respiratory rate at birth;

- Presence of neonatal cephalohaematoma;

- Presence of neonatal illnesses, including feeding difficulties, vomiting, failure to thrive, haemorrhages, pyrexia, septicaemia and other;

- Presence of neonatal breathing difficulties, including Respiratory Distress Syndrome (RDS), intercostal rib recession, grunting, groaning, respiratory infection, apnoeic attacks and other.

## Outcome

Multimorbidity was defined using the National Institute for Health and Care Excellence (NICE) guidelines as the presence of two or more chronic LTCs within a single individual, where at least one is a physical health issue [40]. Due to consistent data collection about these conditions across the adult waves, the following 10 self-reported LTCs were included: asthma; diabetes; epilepsy; chronic back pain; any cancer; auditory issues; hypertension; migraine; eczema and depression. Individuals were categorised into two groups: multimorbidity present (≥ 2 LTCs) or absent (0 or 1 LTC). The prevalence of multimorbidity was calculated at ages 34, 38, 42 and 46.

## Confounders

A Directed Acyclic Graph (DAG) was created using DAGitty v3.0. The DAG enabled a parsimonious approach towards the variables that were included in the final adjusted models

(Fig 1) [41]. Parental confounders, measured at birth, included paternal social class (non-manual workers or manual workers/unpartnered mothers), employment (yes or no/ unpartnered mothers), maternal smoking status (never-/ex-/current), parity (0/1/2/3/4+), and maternal age at birth.

## Mediators

Previous studies have demonstrated associations between adverse neonatal events and cigarette smoking [42], and underweight or elevated Body Mass Index (BMI) in adulthood [43]. The relationships between smoking, abnormal BMI and adult multimorbidity are well established [44,45]. The following covariates, recorded in self-reported questionnaires, were therefore included as potential mediators: smoking (never/ ex-/ current); and BMI(<18.5 kg/ m²/18.5 to <25.0 kg/m²/25.0 to <30 kg/m²/ ≥30.0 kg/m²). This data was gathered at each individual sweep; however, at age 38, BMI data was unavailable due to this sweep having been conducted as a telephone interview.

## Statistical analysis

Data were analysed using STATA v17.0 [46].

Mixed Components Analysis (MCA) was performed to determine the components of the adverse neonatal events score [47]. The number of MCA factors retained was based on the inclusion of components that cumulatively contributed to at least 80% of the dataset's variance [48]. Variables within each retained MCA component with factor loading values greater than 0.3 were included for subsequent analysis (S3 Table) [49].

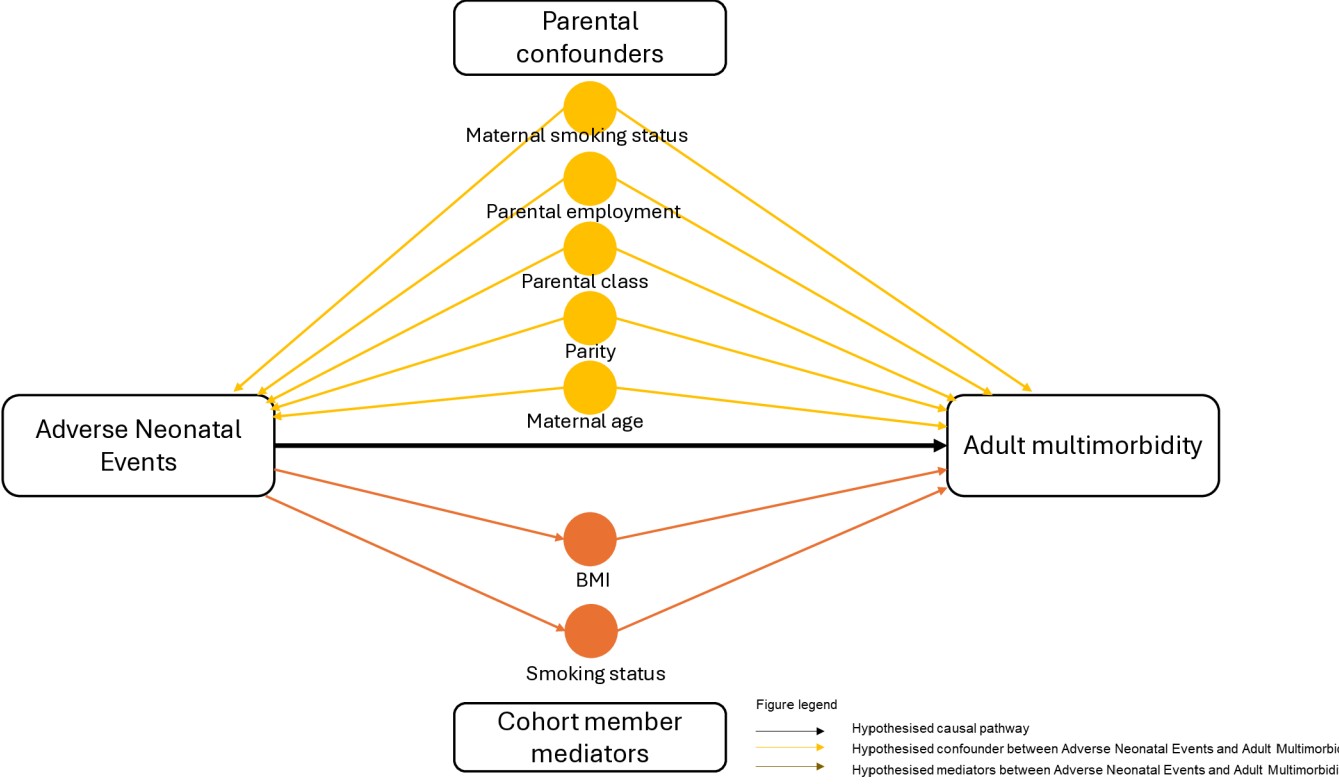

**Fig 1. Causal diagram demonstrating included confounders and mediators following DAG.**

Following MCA, retained variables were combined into composite scores. All categorical variables were transformed into dichotomous outcomes by dummy variable conversion. The retained variable birthweight was categorised into: Low Birth Weight (LBW), defined as less than 2500 grammes [50] and given a score of 1, and all other weights, 0. This was done to allow the computation of an adverse neonatal event score. *Adverse neonatal event* scores ranged from 0 to 6, with 0 implying absence of any adverse neonatal events.

To examine the association between multimorbidity and the adverse neonatal events score, the latter was grouped into 0 (none), 1 (single), or ≥ 2 (multiple). Univariable log-binomial regression models quantified the unadjusted association between adverse neonatal event categories, and the relative risk of developing multimorbidity, at the four adult sweeps individually (ages 34, 38, 42 and 46). Relative risk ratios were estimated using the STATA command 'glm', where the relative risks were estimated from a log-binomial regression model.

The relationship between adverse neonatal events and adult multimorbidity was further explored with inclusion of parental confounders. Paternal social class (non-manual or professional occupations/manual and unskilled occupations which represents a proxy measure of the father's employment type) [51], paternal employment (employed/unemployed) [51] maternal smoking status (Non-smoker or stopped smoking pre-pregnancy/smoked during pregnancy) [52], parity (0/2/3/4+) and maternal age are associated with an increased risk of the exposure of multiple adverse pregnancy outcomes, as well as the outcome of offspring multimorbidity [53,54].

Cohort member mediators included within analyses were BMI group (BMI below 18.5/18.5-24.9/25.0-29.9/BMI over 30) and adult smoking status (never smoked/previous smoker/occasional smoker/smoker), measured at each adult sweep.

Log-binomial regression analysis results are displayed via different models: M0 represents the unadjusted model; M1, M2, M3, M4, M5 are adjusted for paternal class, paternal employment, maternal smoking, maternal age, and parity respectively; M6 includes all parental confounders; M7 accounts for cohort member smoking status; and finally, M8 includes parental covariates, cohort member smoking and BMI status.

## Results

Table 1 shows the sample size and characteristics of the cohort who were present at each sweep. By age 46, 37.4% of the original birth cohort remained in the study. Loss to follow up was significantly higher among males, those who had a low paternal social class, whose fathers were unemployed and whose mothers smoked during pregnancy and/or at birth. However, the proportion of respondents who experienced one or more adverse neonatal events remained similar across the observed samples:13.7% of those who took part in the original birth sweep; 12.6% amongst the age 46 sweep (S3 Table). Multimorbidity was present in 14.6% at age 34; 15.1% at age 38; 21.8% at age 42; and 25.5% at age 46.

For the MCA, data were analysed for 13,371 liveborn, singleton infants, without known congenital anomalies, and after the exclusion of participants missing at least one of the considered adverse neonatal event variable.

MCA was conducted on nine indicators of adverse neonatal events. The first dimension described 88.4% of the sample's total variation. This resulted in the retention of 6 variables ('Birthweight'; 'Neonatal cyanosis'; 'Neonatal cerebral signs'; 'Neonatal illnesses'; 'Neonatal breathing difficulties'; and 'Prolonged duration to establishment of respiratory rate at birth'), which individually contributed to a significant loading weight greater than 0.3 on the factorial axis of Dimension 1. The score value was 0 for 86.3% of the sample, 1 for 10.7%, and ≥ 2 for the remaining 3.0% (Table 1).

**Table 1. Cohort demographics by data sweep.**

| Data sweeps | | | | | |
|---|---|---|---|---|---|
| | **Birth**<br>**n(%)** | **Age 34**<br>**n(%)** | **Age 38**<br>**n(%)** | **Age 42**<br>**n(%)** | **Age 46**<br>**n(%)** |
| **Gender** | | | | | |
| Male | 7,136 (51.6) | 3,455 (47.5) | 3,144 (47.2) | 3,496 (47.9) | 3,091 (48.1) |
| Female | 6,705 (48.4) | 3,820 (52.5) | 3,522 (52.8) | 3,803 (52.1) | 3,338 (51.9) |
| **Adverse Neonatal Events[a]** | | | | | |
| None(0) | 11,936 (86.3) | 6,388 (87.1) | 5,813 (87.2) | 6,352 (87.0) | 5,622 (87.5) |
| Single(1) | 1,487 (10.7) | 733 (10.1) | 682 (10.2) | 758 (10.4) | 649 (10.1) |
| Multiple(≥2) | 419 (3.0) | 204 (2.8) | 171 (2.6) | 189 (2.6) | 158 (2.5) |
| **Paternal social class** | | | | | |
| High(0) | 3,787 (28.2) | 2,303 (31.7) | 2,213 (33.2) | 2,337 (32.0) | 2,139 (33.3) |
| Low(1) | 9,628 (71.8) | 4,972 (68.3) | 4,453 (66.8) | 4,962 (68.0) | 4,290 (66.7) |
| **Paternal employment** | | | | | |
| Yes(0) | 12,412 (92.5) | 6,867 (94.4) | 6,309 (94.6) | 6,874 (94.2) | 6,065 (94.3) |
| No(1) | 1,003 (7.5) | 408 (5.6) | 357 (5.4) | 425 (5.8) | 364 (5.7) |
| **Maternal smoking status** | | | | | |
| Never(0) | 5,694 (42.6) | 3,194 (43.9) | 3,000 (45.0) | 3,229 (44.2) | 2,868 (44.6) |
| Ex(1) | 2,237 (16.7) | 1,310 (18.0) | 1,180 (17.7) | 1,283 (17.6) | 1,134 (17.7) |
| Current(2) | 5,440 (40.7) | 2,771 (38.1) | 2,486 (37.3) | 2,787 (38.2) | 2,427 (37.7) |
| **Parity** | | | | | |
| 0 | 5,100 (37.8) | 2,782 (39.4) | 2,662 (39.6) | 2,582 (39.2) | 2,448 (40.5) |
| 1 | 4,389 (32.5) | 2,413 (34.2) | 2,346 (34.9) | 2,247 (34.1) | 2,050 (33.9) |
| 2 | 2,159 (16.0) | 1,078 (15.3) | 1,028 (15.3) | 1,023 (15.5) | 910 (15.0) |
| 3 | 974 (7.2) | 52 (6.4) | 392 (5.8) | 408 (6.2) | 365 (6.0) |
| 4 | 885 (6.6) | 341 (4.8) | 292 (4.4) | 324 (4.9) | 277 (4.6) |
| **Maternal age: mean (SD)** | 26.0 (5.4) | 26.0 (5.3) | 26.0 (5.2) | 26.0 (5.3) | 26.0 (5.3) |
| **Multimorbidity[b]** | | | | | |
| No | – | 6,215 (85.4) | 5,557 (84.9) | 5,704 (78.2) | 4,749 (74.5) |
| Yes | – | 1,060 (14.6) | 1,009 (15.1) | 1,595 (21.8) | 1,630 (25.5) |
| **Smoking** | | | | | |
| Never | – | 3,923 (45.3) | 3,151 (47.3) | 3,386 (46.4) | 3,061 (48.0) |
| Ex | – | 1,737 (23.9) | 1,836 (27.5) | 2,057 (28.2) | 2,036 (31.9) |
| Current | – | 2,233 (30.7) | 1,679 (25.2) | 1,854 (25.4) | 1,282 (20.1) |
| **BMI** | | | | | |
| <18.5 | – | 97 (1.38) | Ŧ | 79 (1.2) | 36 (0.65) |
| 18.5-24.9 | – | 3,381 (47.9) | | 2,640 (40.1) | 1,571 (28.5) |
| 25-29.9 | – | 2,403 (34.1) | | 2,428 (36.9) | 2,051 (37.2) |
| ≥30.0 | | 1,172 (16.6) | | 1,437 (21.8) | 1,861 (33.7) |

[a]Composite score of characteristics- Delay in time to establish regular respiration; Cyanosis; Cerebral signs; Breathing difficulties; Other illnesses; birthweight.

[b]Presence of 2 or more Long Term Conditions.

BMI = Body Mass Index (Weight(kg)/Height$^2$ (metres).

ŦBMI data not available for age 38 sweep, as this was a telephone interview.

There were no significant associations between experiencing one or more adverse neonatal events at birth and multimorbidity at ages 34, 42 and 46 (Table 2). In the age 38 data sweep, having two or more adverse neonatal events was associated with a greater risk of adult multimorbidity, despite adjustment for parental confounders and cohort member smoking status (RR 1.41; 95% CI 1.05 – 1.88).

## Discussion

### Main findings

This study explored the associations between experiencing multiple adverse neonatal events at birth and subsequent multimorbidity at multiple time points during adulthood. At 38 years, those who had experienced two or more adverse neonatal events at birth, had a 41% higher risk of multimorbidity compared to those who had no history of adverse neonatal events. This association was maintained in the adjusted models. This could suggest that any effect of the *adverse neonatal events* score on adult multimorbidity is through routes other than the covariates included in the models.

At the three other age sweeps, the associations between adverse neonatal events at birth and mid-life multimorbidity were not statistically significant. Although as hypothesised, the general direction of risk suggests that the more adverse neonatal events at birth, the higher the risk of multimorbidity.

Studies that have considered combined neonatal events and multimorbidity are lacking, making comparisons to previous research difficult. However, given we found an association between adverse neonatal events and adult multimorbidity at one timepoint only (age 38), our study can only go some way to support previous research that have found single neonatal events such as birthweight [14,25], maternal age [15] and maternal BMI [15] to be associated with future multimorbidity risk in adulthood. In additional, our research provides little support to other research that have found single neonatal adversity such as low birth weight and preterm births to be associated to single adult LTCs including cardiovascular disease [21] and mid-life diabetes [21]. However, it is important to note that the pathways between neonatal adversity and single LTCs compared to multimorbidity may differ, and so caution must be taken when making comparisons between research.

### Strengths and limitations

This is the first study, to our knowledge, which explores the association between exposure to multiple adverse neonatal events at birth, and later-life risk of multimorbidity. A strength of this study is the utilisation of a large cohort study data at multiple time-points in an individual's life course. The longitudinal nature of the study means that temporality is established, facilitating a life course interpretation. This study also considered the impact of various intergenerational and individual social determinants on future health; this data may not be readily available in other primary or secondary care datasets.

This study comprised of individuals born over 50 years ago. There have been numerous changes to obstetric, and neonatal care practices during this time [55]. Examples of this include the ascending trend of iatrogenic, and idiopathic preterm births; and greater frequency of maternal morbidity, such as Gestational Diabetes Mellitus (GDM), which impacts birthweight [56,57]. Caution must therefore be exercised in comparing the adverse neonatal events of 1970 to the present day. In the BCS70, over 99% of the cohort identified as White British. Therefore, the established associations between ethnic minority status and increased risk of multimorbidity, particularly at earlier age groups could not be explored in this study [58]. Self-reported rather than measured conditions were used to calculate multimorbidity, which could have also introduced problems with data reliability [59].

**Table 2. Models exploring associations between adverse neonatal events and multimorbidity.**

| | Analytical Sample | Model | RR | Multimorbidity 95% CI |
|---|---|---|---|---|
| **Age 34** | 7,066 | | | |
| *0 adverse neonatal events* | | | Ref | |
| *1 adverse neonatal event* | | M0 | 0.93 | 0.77 – 1.13 |
| | | M1 | 0.93 | 0.76 – 1.13 |
| | | M2 | 0.93 | 0.76 – 1.12 |
| | | M3 | 0.92 | 0.76 – 1.12 |
| | | M4 | 0.93 | 0.77 – 1.13 |
| | | M5 | 0.93 | 0.76 – 1.13 |
| | | M6 | 0.92 | 0.75 – 1.11 |
| | | M7 | 0.91 | 0.75 – 1.11 |
| | | M8 | 0.92 | 0.76 – 1.11 |
| *≥2 adverse neonatal events* | | M0 | 1.11 | 0.81 – 1.54 |
| | | M1 | 1.12 | 0.81 – 1.54 |
| | | M2 | 1.12 | 0.81 – 1.54 |
| | | M3 | 1.10 | 0.80 – 1.52 |
| | | M4 | 1.12 | 0.81 – 1.54 |
| | | M5 | 1.10 | 0.80 – 1.52 |
| | | M6 | 1.09 | 0.79 – 1.51 |
| | | M7 | 1.11 | 0.80 – 1.53 |
| | | M8 | 1.10 | 0.80 – 1.51 |
| **Age 38** | 6,687 | | | |
| *0 adverse neonatal event* | | | Ref | |
| *1 adverse neonatal event* | | M0 | 0.98 | 0.81 – 1.18 |
| | | M1 | 0.97 | 0.80 – 1.17 |
| | | M2 | 0.97 | 0.81 – 1.17 |
| | | M3 | 0.97 | 0.80 – 1.17 |
| | | M4 | 0.98 | 0.81 – 1.18 |
| | | M5 | 0.97 | 0.81 – 1.18 |
| | | M6 | 0.97 | 0.80 – 1.17 |
| | | M7 | 0.96 | 0.80 – 1.16 |
| | | M8 | | Unavailable |
| *≥2 adverse neonatal events* | | **M0** | **1.38** | **1.03 – 1.85** |
| | | **M1** | **1.38** | **1.03 – 1.85** |
| | | **M2** | **1.38** | **1.03 – 1.85** |
| | | **M3** | **1.37** | **1.02 – 1.83** |
| | | **M4** | **1.38** | **1.03 – 1.85** |
| | | **M5** | **1.38** | **1.03 – 1.86** |
| | | **M6** | **1.37** | **1.03 – 1.84** |
| | | **M7** | **1.41** | **1.05 – 1.88** |
| | | M8 | | Unavailable |
| **Age 42** | 6,584 | | | |
| *0 adverse neonatal events* | | | Ref | |
| *1 adverse neonatal event* | | M0 | 0.99 | 0.86 – 1.16 |
| | | M1 | 0.99 | 0.86 – 1.16 |
| | | M2 | 1.00 | 0.86 – 1.16 |
| | | M3 | 0.99 | 0.85 – 1.15 |
| | | M4 | 0.99 | 0.86 – 1.16 |
| | | M5 | 0.99 | 0.85 – 1.15 |
| | | M6 | 0.98 | 0.85 – 1.15 |

*(Continued)*

**Table 2.** (Continued)

| | Analytical Sample | Model | RR | Multimorbidity 95% CI |
|---|---|---|---|---|
| | | M7 | 0.99 | 0.85 – 1.15 |
| | | M8 | 1.00 | 0.86 – 1.16 |
| ≥2 adverse neonatal events | | M0 | 1.29 | 1.00 – 1.64 |
| | | M1 | 1.28 | 1.00 – 1.64 |
| | | M2 | 1.29 | 1.00 – 1.64 |
| | | M3 | 1.27 | 0.99 – 1.62 |
| | | M4 | 1.29 | 1.00 – 1.64 |
| | | M5 | 1.28 | 1.00 – 1.63 |
| | | M6 | 1.27 | 0.99 – 1.62 |
| | | M7 | 1.27 | 0.99 – 1.62 |
| | | M8 | 1.24 | 0.98 – 1.58 |
| **Age 46** | *6,030* | | | |
| *0 adverse neonatal events* | | | Ref | |
| *1 adverse neonatal event* | | M0 | 0.98 | 0.84 – 1.13 |
| | | M1 | 0.97 | 0.84 – 1.12 |
| | | M2 | 0.97 | 0.84 – 1.13 |
| | | M3 | 0.97 | 0.83 – 1.11 |
| | | M0 | 0.98 | 0.84 – 1.12 |
| | | M5 | 0.97 | 0.84 – 1.13 |
| | | M6 | 0.97 | 0.83 – 1.12 |
| | | M7 | 0.96 | 0.83 – 1.11 |
| | | M8 | 0.96 | 0.83 – 1.11 |
| *≥2 adverse neonatal events* | | M0 | 0.92 | 0.69 – 1.23 |
| | | M1 | 0.93 | 0.69 – 1.23 |
| | | M2 | 0.92 | 0.69 – 1.23 |
| | | M3 | 0.91 | 0.68 – 1.22 |
| | | M0 | 0.92 | 0.69 – 1.23 |
| | | M5 | 0.92 | 0.69 – 1.24 |
| | | M6 | 0.92 | 0.69 – 1.23 |
| | | M7 | 0.91 | 0.68 – 1.21 |
| | | M8 | 0.92 | 0.69 – 1.22 |

M0: Unadjusted model.

M1:Adjusts for paternal class.

M2: Adjusts for paternal employment.

M3: Adjusts for maternal smoking.

M4: Adjust for maternal age.

M5: Adjust for parity.

M6: M1 + M2 + M3 + M4 + M5.

M7: M6 + cohort member smoking status.

M8: M7 + cohort member BMI.

*Ref*: Reference group; *n* = number of participants included.

significant at <0.05 level highlighted in bold.

Data availability limited the number of self-reported conditions we could consider and there are a number of common diseases missing from our outcome such as cardiovascular diseases, rheumatoid diseases, and some psychiatric diseases. It is therefore likely that we are underestimating the prevalence of multimorbidity amongst the cohort. Data availability also

precluded the opportunity to considered other potential adult mediators including physical activity and nutrition/diet as they were not repeatedly recorded across all the adult sweeps.

Additionally, no differentiation was made between the material impact of different diseases or disease severity to the individual. Similarly, all adverse neonatal events were given the same weighting. No BMI data were available for age 38. Although several confounding and mediating variables were adjusted for in analysis, it was beyond the scope of this paper to fully explore the pathways linking adverse neonatal events to later-life multimorbidity.

By age 46, only 37% of the original cohort remained. Missing observations were managed by omission in this study. Loss to follow up and missing data may have led to a reduction in statistical power, lowered representativeness of data and introduced bias [60].

### Interpretation

There are several possible reasons for the observed and unobserved associations within this study. One potential explanation is that adverse neonatal events exert a differential impact on multimorbidity with increasing age. The peak impact of adverse neonatal events on multimorbidity is subsequently observed at age 38, beyond which, other risk factors, like smoking and abnormal BMI, perhaps demonstrate a more significant effect, weakening the association with poor birth outcomes. It can be hypothesised that at age 38, the full impact of adult behavioural choices on multimorbidity is yet to be realised. Indeed, in this cohort, the unadjusted relative risk of multimorbidity increased with each subsequent sweep, for those with a BMI greater than/ equal to $30 \, kg/m^2$ (S4 Table). Although current smoking was associated with multimorbidity throughout all adult sweeps, ex-smoking was not a significant determinant of multimorbidity until age 42, which may demonstrate the dose-dependent nature of 'smoked pack years' on the risk of multiple LTCs (S4 Table). This is supported by previous literature, which demonstrated that multimorbidity was associated with childhood Adverse Childhood Experiences (ACEs) at middle-age, but not at older age groups [61].

Another explanation is that a Type II error has occurred, i.e., the null hypothesis that there is no association between adverse neonatal events at birth and adult multimorbidity at ages 34, 42 and 46 has been falsely accepted given the smaller sample size of adverse neonatal events. It may be surmised that because of a greater differential attrition amongst those with multimorbidity, the true prevalence of multimorbidity has been underestimated, and the association between adverse neonatal events and adult multimorbidity was diluted [62].

An important characteristic that has not been addressed due to lack of access to relevant data, is the mortality rate. Previous literature has established the link between some adverse neonatal events including early term and preterm birth and premature mortality [63,64]. In a landmark study, men with the lowest recorded birthweights had the highest death rates from ischaemic heart disease [65]. It is therefore plausible that individuals who experienced two or more adverse neonatal events did in fact have a higher prevalence of multimorbidity but were more likely to die before follow-up, introducing further bias.

The impact of single adverse childhood events on the adult incidence of multimorbidity has been explored previously [23,24]. In one of these studies, childhood neglect exhibited an increased risk of adult multimorbidity of similar magnitude to established risk factors such as smoking and obesity [66]. In the BCS70 cohort, an increased relative risk of multimorbidity persisted at age 38 for those who had experienced two or more adverse neonatal events, despite adjustment for cohort member smoking status. This adds to the body of evidence that poor early-life factors could be as predictive of ill health in adulthood as unhealthy behavioural factors.

## Conclusion

In this analysis, adverse neonatal events at birth demonstrated an independent detrimental effect on multimorbidity at age 38, and represent a potential determinant of midlife multimorbidity. Based on the results presented here it is important that adverse neonatal events continue to be taken into account when considering how to tackle the growing public health burden of multimorbidity.

## Supporting information

**S1 Fig. A flow chart highlighting the analytical sample at each sweep.**
(TIF)

**S1 Table. List of congenital abnormalities.**
(DOCX)

**S2 Table. Study variables.**
(DOCX)

**S3 Table. Structure of the first axis of the MCA conducted with 9 adverse neonatal indicators.**
(DOCX)

**S4 Table. Unadjusted association between multimorbidity and cohort member BMI and smoking status at ages 34, 38, 42 and 46.**
(DOCX)

## Acknowledgements

We would like to thank all those who were part of the BCS70 cohort study, without whose participation this analysis would not have been possible.

## Author contributions

**Conceptualization:** Jeeva John, Seb Stannard, Simon D. S. Fraser, Ann Berrington, Nisreen A. Alwan.

**Data curation:** Jeeva John.

**Formal analysis:** Jeeva John, Seb Stannard.

**Funding acquisition:** Seb Stannard.

**Investigation:** Jeeva John.

**Methodology:** Jeeva John.

**Project administration:** Jeeva John.

**Software:** Jeeva John.

**Supervision:** Seb Stannard, Nisreen A. Alwan.

**Validation:** Jeeva John.

**Visualization:** Jeeva John.

**Writing – original draft:** Jeeva John.

**Writing – review & editing:** Jeeva John, Seb Stannard, Simon D. S. Fraser, Ann Berrington, Nisreen A. Alwan.

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
