## [Decision Letter · Decision Letter 0]

17 Sep 2024

PONE-D-24-17447Clusters and associations of adverse neonatal events with adult risk of multimorbidity: A secondary analysis of birth cohort data.PLOS ONE

Dear Dr. Macmillan-John,

Thank you for submitting your manuscript to PLOS ONE. After careful consideration, we feel that it has merit but does not fully meet PLOS ONE’s publication criteria as it currently stands. Therefore, we invite you to submit a revised version of the manuscript that addresses the points raised during the review process. Please submit your revised manuscript by Nov 01 2024 11:59PM. If you will need more time than this to complete your revisions, please reply to this message or contact the journal office at plosone@plos.org . Please include the following items when submitting your revised manuscript:

We look forward to receiving your revised manuscript.

Kind regards,

Khin Thet Wai, MBBS, MPH, MA

Academic Editor

PLOS ONE

Journal Requirements:

For additional information about PLOS ONE ethical requirements for human subjects research, please refer to http://journals.plos.org/plosone/s/submission-guidelines#loc-human-subjects-research .

“JJ received no funds, grants, or other support for this project. SS, SF, AB and NA received financial support for this research from National Institute for Health Research (NIHR) Artificial Intelligence for Multiple and Long-Term Conditions (NIHR203988).”

5. Please note that funding information should not appear in the Acknowledgments section or other areas of your manuscript. We will only publish funding information present in the Funding Statement section of the online submission form. Please remove any funding-related text from the manuscript.

6. Please note that your Data Availability Statement is currently missing the DOI/accession number of each dataset or a direct link to access each database. If your manuscript is accepted for publication, you will be asked to provide these details on a very short timeline. We therefore suggest that you provide this information now, though we will not hold up the peer review process if you are unable.

7. Your ethics statement should only appear in the Methods section of your manuscript. If your ethics statement is written in any section besides the Methods, please move it to the Methods section and delete it from any other section. Please ensure that your ethics statement is included in your manuscript, as the ethics statement entered into the online submission form will not be published alongside your manuscript.

8. We notice that your supplementary tables are included in the manuscript file. Please remove them and upload them with the file type 'Supporting Information'. Please ensure that each Supporting Information file has a legend listed in the manuscript after the references list.

Reviewers' comments:

Reviewer's Responses to Questions

**Comments to the Author**

1. Is the manuscript technically sound, and do the data support the conclusions?

Reviewer #1: Partly

Reviewer #2: Partly

2. Has the statistical analysis been performed appropriately and rigorously? 

Reviewer #1: No

Reviewer #2: Yes

3. Have the authors made all data underlying the findings in their manuscript fully available?

Reviewer #1: Yes

Reviewer #2: Yes

4. Is the manuscript presented in an intelligible fashion and written in standard English?

Reviewer #1: Yes

Reviewer #2: Yes

5. Review Comments to the Author

Reviewer #1: This study defined clusters of adverse neonatal events and assessed their association with multimorbidity in middle age. The topic of the article is interesting and the article is clearly written. However, I see some possibilities for improving the quality of the article by presenting and discussing both adverse neonatal events and multimorbidity, and the potential pathways that link these together in a more systematic and thorough way. Below I present some detailed major and minor comments.

Abstract

•It would be good to list the adverse neonatal events in the abstract or at least give some examples of them.

•The conclusion is about prevention of adverse neonatal events, but prevention is not the focus of this article.

Introduction

•Introduction in general: At times the text is presented in a rather superficial level and is slightly off the focus of the topic of research, it lacks depth.

•I would have wished to see more introduction to the different forms of multimorbidity (specific disease clusters, physical and mental multimorbidity).

•Adverse neonatal events are introduced very superficially, only preterm birth is mentioned. This is important for the reader to understand why specific adverse neonatal events were chosen to be studied. In general, one must read the manuscript very far until methods to find out how the authors define adverse neonatal events.

•The introduction lacks discussion on possible mechanisms by which neonatal adverse events could cause multimorbidity.

Methods

•Study design and population: please specify that Appendix A presents the list of congenital abnormalities excluded from the study population.

•Exposures: what is the justification for selecting these possible exposures for creating the score? They seem to be on a very different level, for example cephalohematoma is maybe just a proxy of the mode of delivery whereas preterm birth and LBW are shown in many studies to have a profound impact on later health.

•Exposures: why did the authors decide to use low birthweight instead of small for gestational age (SGA) when the gestational age is known? LBW and preterm birth are largely overlapping.

•Exposures: How were the exposures measured and how reliable is the assessment?

•Outcome: Multimorbidity is stated as the outcome and defined as the presence of two or more chronic conditions. However, the results also present one chronic condition as one of the outcomes. Could the authors justify this?

•Outcome: apparently the outcomes were selected based on data availability. There are some common diseases missing, such as cardiovascular diseases, rheumatoid diseases, many psychiatric diseases, COPD, to name a few. Lack of these should at least be discussed as a limitation in the discussion.

•Confounders and mediators: how were the confounders and mediators measured?

•Confounders: did the authors consider adding parental health status, maternal age or parity as a confounder?

•Mediators: did the authors consider adding other health behaviors, such as physical activity or nutrition as mediators?

•Statistical analysis: the list of adverse neonatal events is not clear. 6 events are mentioned in the text, but the list includes only 5 in between the semicolons. Furthermore, it seems that some of the events presented earlier as individual events are now combined in a group? This is rather confusing.

•Statistical analysis and throughout the text: how did the authors get a relative risk from logistic regression (which gives odds ratios)?

•Statistical analysis: given that the outcome is very common (up to 25%), logistic regression might give inflated odds ratios compared to relative risk. Did the authors consider using other statistical methods instead of logistic regression?

•Statistical analysis: the additional explanation of paternal social class (page 9, lines 191-193) would be better suited earlier in methods where this confounder is introduced.

Results

•The first aim is to identify and characterize clustering of adverse neonatal events. However, this clustering is not presented in the results. It would be very useful for the reader to understand the exposure better: how common were the individual adverse events, and what were the most common clusters? Did the observed associations originate mainly from preterm birth / low birthweight?

•In general, it is difficult for the reader to understand the different numbers of participants that are presented in the results and at the beginning of the methods. I believe the manuscript would benefit from a proper flow chart, starting from the beginning of the study (exclusions presented in Methods / study population). Further, it is difficult to understand the differences in numbers of participants between Table 1 and Table 2.

•Appendix B could have been introduced earlier in Methods/Statistical analyses

•Page 11, lines 225-227 gives yet another list of adverse neonatal events. Please clarify how this is different from the one presented in Methods / statistical analysis.

•The supplementary table presents the associations between BMI and smoking and multimorbidity. It would be also interesting to see the association between the individual neonatal adverse events and multimorbidity.

Discussion

•Strengths: can the sample be considered nationally representative after the loss to follow-up?

•Strengths: long duration of follow-up until middle age is definitely a strength of this study.

•The discussion is lacking comparing the results of this study to previous studies (the studies compared to are either about mortality as the outcome or adverse childhood events as an exposure). There might not be many studies studying the risk of multimorbidity in people with several neonatal adverse events, but at least there are some for preterm birth and maybe other exposures individually (where the outcome is measured at slightly younger age). Similarly, there is a large body of literature assessing the associations between single neonatal adverse events (such as preterm birth) and single diseases, not just mortality which is mentioned in the discussion.

•"Adverse neonatal events" mentioned in page 15, lines 310-311: the referred articles only study early term and preterm birth. It would be good to specify this.

•Lines 324-327 are rather vague and theoretical, and not in the focus of this study. Please consider removing them.

Conclusion

•Interventions to prevent adverse neonatal events are not in the focus of this study. Please consider rewording the conclusion. For example, based on this study we could suggest that early life factors should be better taken into account in the adult health care.

Minor comments

•The language in the article is fluent, but there are some typos, e.g. data are plural.

Reviewer #2: General comment

The manuscript is interesting and highlights the association between the neonatal adversity and adulthood multimorbidity. The research objectives have been partially achieved according to the statistical analysis applied and the findings. The strength and limitation of study are well explained.

Specific comments

I would like to clarify some issues and comment about statistical analysis.

To determine the components of the adverse neonatal event score, I am wondering whether the authors used mixed component analysis or principal component analysis for mixed data. These two methods are not synonymous.

The authors presented relative risk (RR) of developing adulthood multimorbidity for subjects with one or ≥2 adverse neonatal events, compared with no adverse neonatal events, using logistic regression. I would like to know about how the RR was estimated from odds ratio produced by the logistic regression model.

Another comment is related to model 5 (M 5) and model 6 (M 6), where the authors mentioned that cohort member smoking status and BMI were considered the mediators (lines 194 and 195). A logistic regression model can adjust for confounders but not for mediators, as this model estimates the direct effect of independent variables or covariates. For assessment of mediators or mediation effect, both the direct effect and the indirect effect need to be measured. This requires mediation analysis, as logistic regression alone cannot distinguish between these effects. Therefore, if you would like to assess the mediating role of cohort member smoking and BMI status, mediation analysis is required, as a logistic regression model cannot provide separate estimates of the direct and indirect effects.

There is also inconsistency in expressing the characteristics or variables for defining the composite score of adverse neonatal events (i.e., six retained adverse neonatal variables) in lines 178-181, 213-214, and then 225-227. These phrases need to be revised for clarity and consistency.

6. PLOS authors have the option to publish the peer review history of their article (what does this mean? ). If published, this will include your full peer review and any attached files.

**Do you want your identity to be public for this peer review?** For information about this choice, including consent withdrawal, please see our Privacy Policy .

Reviewer #1: No

Reviewer #2: No

---

## [Author Response · Author response to Decision Letter 1]

23 Dec 2024

Journal Requirements:

Thank you, we have formatted the paper following the guidelines provided.

3. If you are reporting a retrospective study of medical records or archived samples, please ensure that you have discussed whether all data were fully anonymized before you accessed them and/or whether the IRB or ethics committee waived the requirement for informed consent. If patients provided informed written consent to have data from their medical records used in research, please include this information. Once you have amended this/these statement(s) in the Methods section of the manuscript, please add the same text to the “Ethics Statement” field of the submission form (via “Edit Submission”). For additional information about PLOS ONE ethical requirements for human subjects research, please refer to http://journals.plos.org/plosone/s/submission-guidelines#loc-human-subjects-research.

We have added the following sentence to address point 2 and 3:

‘All data were fully anonymized prior to data access. Participants provided fully informed consent, full details of the consent process are outlined by the Centre for Longitudinal Studies and can be found elsewhere. (https://cls.ucl.ac.uk/wp-content/uploads/2017/07/BCS70-Ethical-review-and-Consent-2019.pdf)’

4. We note that the grant information you provided in the ‘Funding Information’ and ‘Financial Disclosure’ sections do not match. When you resubmit, please ensure that you provide the correct grant numbers for the awards you received for your study in the ‘Funding Information’ section.

5. Thank you for stating the following financial disclosure: “JJ received no funds, grants, or other support for this project. SS, SF, AB and NA received financial support for this research from National Institute for Health Research (NIHR) Artificial Intelligence for Multiple and Long-Term Conditions (NIHR203988).” Please state what role the funders took in the study.  If the funders had no role, please state: "The funders had no role in study design, data collection and analysis, decision to publish, or preparation of the manuscript." If this statement is not correct you must amend it as needed. Please include this amended Role of Funder statement in your cover letter; we will change the online submission form on your behalf.

The funding statement now states:

‘This research forms part of the MELD-B project supported by the National Institute for Health Research (NIHR) Artificial Intelligence for Multiple and Long-Term Conditions (NIHR203988). The views expressed are those of the authors and not necessarily those of the NIHR or the Department of Health and Social Care. The funders had no role in study design, data collection and analysis, decision to publish, or preparation of the manuscript.’

6. Please note that funding information should not appear in the Acknowledgments section or other areas of your manuscript. We will only publish funding information present in the Funding Statement section of the online submission form. Please remove any funding-related text from the manuscript.

We have removed the funding statement from acknowledgement.

7. Please note that your Data Availability Statement is currently missing the DOI/accession number of each dataset or a direct link to access each database. If your manuscript is accepted for publication, you will be asked to provide these details on a very short timeline. We therefore suggest that you provide this information now, though we will not hold up the peer review process if you are unable.

Thank you. We have included the following statement at the end of the manuscript: ‘The BCS70 datasets generated and analysed in the current study are available through the UK Data Archive repository (available here: http://www.cls.ioe.ac.uk/page.aspx?&sitesectionid=795).’

8. Your ethics statement should only appear in the Methods section of your manuscript. If your ethics statement is written in any section besides the Methods, please move it to the Methods section and delete it from any other section. Please ensure that your ethics statement is included in your manuscript, as the ethics statement entered into the online submission form will not be published alongside your manuscript.

We have moved the ethics statement to the methods.

9. We notice that your supplementary tables are included in the manuscript file. Please remove them and upload them with the file type 'Supporting Information'. Please ensure that each Supporting Information file has a legend listed in the manuscript after the references list.

Thank you. We have removed the supplementary tables from the main manuscript and have uploaded them individually. We now include a legend at the end of the manuscript after the references.

Review Comments 1

Abstract

It would be good to list the adverse neonatal events in the abstract or at least give some examples of them.

Thank you. We now include some examples of neonatal events in the abstract.

The conclusion is about prevention of adverse neonatal events, but prevention is not the focus of this article.

We have updated the conclusion in the abstract to reflect the fact that this paper was not about prevention.

Introduction

1. Introduction in general: At times the text is presented in a rather superficial level and is slightly off the focus of the topic of research, it lacks depth.

Thank you. In the introduction we now include the following paragraph identifying previous research that has considered the relationship between childhood exposures and multimorbidity. We additionally discuss how the relationship between neonatal adversity and multimorbidity has not been explored:

‘Research has demonstrated that certain early-life characteristics are associated with multimorbidity in adulthood [14-22]. Amongst the 1970 British Cohort Study, variables in childhood including parental social class, birthweight, childhood BMI, cognitive ability and behaviour were associated to a count of multimorbidity at midlife [14]. For the Hertfordshire cohort study, higher rates of childhood illnesses were associated with future multimorbidity and higher medication counts at ages 64-68 [15]. Amongst a birth cohort born in Helsinki, individuals born to mothers under the age of 25, mothers with a BMI above 25, individuals who had a birthweight less than 2.5kg, those with rapid growth in height and weight from birth until age 11, wartime separation and paternal occupational class were all associated with a faster rate of chronic disease accumulation from midlife onwards [16]. Other research has found that early childhood conditions including parental socioeconomic status [17-22], poor childhood health [17,21], child maltreatment [22], child adversity including abuse and neglect [19], negative caregiver’s characteristics [19], food restriction [21], child labour [21], and stressful life events [21], were associated with multimorbidity across the adult life course. However, with the exception of birthweight and maternal age, previous research has yet to explore the relationship between neonatal adversity and multimorbidity.’

In addition in response to comment 4, we have added additional information on possible mechanisms by which neonatal adverse events could cause multimorbidity. We believe having addressed both these comments the introduction is now more focused on our research.

2. I would have wished to see more introduction to the different forms of multimorbidity (specific disease clusters, physical and mental multimorbidity).

Thank you for this important point. We do agree that multimorbidity is a highly complex topic. We know that certain long-term conditions cluster together, and that the sequence that people develop conditions also vary. Multimorbidity analyses would be enhanced by a better understanding of burdensomeness and complexity, and what they mean to patients and carers. We are aware that multimorbidity is commonly defined as having two or more LTCs (as was done in this paper), but there is a clear need to move away from LTC counts towards a more sophisticated understanding of multimorbidity, considering the interplay between wider social determinants and disease, the influence of mental and physical conditions, and the importance of disease stage/severity.

However, given the data restriction of our research, and our small sample size of participants, we were only able to consider a broad definition of multimorbidity (i.e., a long-term condition count), and we therefore felt it was beyond the scope of this paper to introduce the different forms of multimorbidity in the introduction, especially as this would likely draw attention away from the focus of this paper which was predominantly around adverse neonatal events.

3. Adverse neonatal events are introduced very superficially, only preterm birth is mentioned. This is important for the reader to understand why specific adverse neonatal events were chosen to be studied. In general, one must read the manuscript very far until methods to find out how the authors define adverse neonatal events.

Thank you, we now include how we define neonatal events in the introduction (line 111-114), as well as in the methods section.

4. The introduction lacks discussion on possible mechanisms by which neonatal adverse events could cause multimorbidity.

Thank you, we have added the following paragraph to the introduction section on page 5:

‘The effect of early life events on multimorbidity can be broadly explained by two main lifecourse epidemiological paradigms: the “critical period” theory, in which biological imprinting at important time-points, make an individual more susceptible to compromised health in adult life [31] and; the “accumulation of risk” model, which states that cumulative adverse early-life events contribute to poor adult health [32]. Both theories have potential relevance to the aetiology of multimorbidity, and have therefore been considered as a potential mechanism for this study.’

Methods

1. Study design and population: please specify that Appendix A presents the list of congenital abnormalities excluded from the study population.

Thank you, on line 143 we now state: ‘Due to the association between congenital abnormalities and adult multimorbidity, infants with known congenital abnormalities (outlined in S1 Table) were excluded’

2. Exposures: what is the justification for selecting these possible exposures for creating the score? They seem to be on a very different level, for example cephalohematoma is maybe just a proxy of the mode of delivery whereas preterm birth and LBW are shown in many studies to have a profound impact on later health.

Thank you, given the first stage of our analysis involved clustering, we took an inclusive approach (in terms of what was available in the data) to the selection of possible exposures. As such, we selected all variables available within our dataset that the research team felt could potentially be linked to the outcome. We agree that some variables may be proxy for others, however these relationships between variables would have been drawn out during the clustering analysis, as such, we only carried exposures forward based on the results of the clustering analysis.

3. Exposures: why did the authors decide to use low birthweight instead of small for gestational age (SGA) when the gestational age is known? LBW and preterm birth are largely overlapping.

Thank you for raising this important point. We agree that SGA is a more appropriate measure of size at birth if considered as an outcome. However, we are here considering it as a predictor. Also, we initially considered birthweight and gestational age as two independent exposures within the clustering analysis. Based on that, birthweight was retained, and gestational age was not. As such, because gestational age was not retained we made the decision to progress with birthweight in grams.

4. Exposures: How were the exposures measured and how reliable is the assessment?

Thank you, data at birth (exposure) were collected by the midwives who had been present at the birth and, in addition, information was extracted from clinical records. We would suggest that this reflects a reliable assessment of neonatal events. We have updated the exposure section of the methods (p.7) to explain how the exposure were measured.

5. Outcome: Multimorbidity is stated as the outcome and defined as the presence of two or more chronic conditions. However, the results also present one chronic condition as one of the outcomes. Could the authors justify this?

We are unsure about this comment, we have reviewed the manuscript including supplementary materials and we can confirm we only present the results for the outcome of two or more conditions (multimorbidity). We do not present any results in relation to one condition only.

6. Outcome: apparently the outcomes were selected based on data availability. There are some common diseases missing, such as cardiovascular diseases, rheumatoid diseases, many psychiatric diseases, COPD, to name a few. Lack of these should at least be discussed as a limitation in the discussion.

Thank you, this is an important point. We now state in the limitations that:

‘Data availability limited the number of self-reported conditions we could consider and there are a number of common diseases missing for our outcome such as cardiovascular diseases, rheumatoid diseases, and psychiatric diseases. It is therefore likely that we are underestimating the prevalence of multimorbidity amongst the cohort.’

7. Confounders and mediators: how were the confounders and mediators measured?

Thank you, we now include a description of how the confounders and mediators were measured on page 8.

8. Confounders: did the authors consider adding parental health status, maternal age or parity as a confounder?

Thank you. On reflection we agree with the reviewer that some of these factors should be considered as confounders. Unfortunately, parental health status data was not available. However we now include maternal age and parity as confounders in our regression model (Table 2), and we have updated the results accordingly. Overall, the inclusion of both confounders does not substantially changed the results of the paper.

9. Mediators: did the authors consider adding other health behaviors, such as physical activity or nutrition as mediators?

Unfortunately, data availability precluded the opportunity to considered other mediators including physical activity and nutrition/diet as they were not recorded repeatedly across all the adult sweeps. We now discuss this point as a limitation on page 15-16.

10. Statistical analysis: the list of adverse neonatal events is not clear. 6 events are mentioned in the text, but the list includes only 5 in between the semicolons. Furthermore, it seems that some of the events presented earlier as individual events are now combined in a group? This is rather confusing.

Thank you for this important point. Reviewing the paper again we noted that there was inconsistency regarding the six retained variables that defined the composite score. We apologise for this. To confirm, the 6 variables included: ‘birthweight’; ‘Neonatal cyanosis’; ‘Neonatal cerebral signs’; ‘Neonatal illnesses’; ‘Neonatal breat

---

## [Decision Letter · Decision Letter 1]

13 Jan 2025

PONE-D-24-17447R1Clusters and associations of adverse neonatal events with adult risk of multimorbidity: A secondary analysis of birth cohort data.PLOS ONE

Dear Dr. Macmillan-John,

Thank you for submitting your manuscript to PLOS ONE. After careful consideration, we feel that it has merit but does not fully meet PLOS ONE’s publication criteria as it currently stands. Therefore, we invite you to submit a revised version of the manuscript that addresses the points raised during the review process. Please submit your revised manuscript by Feb 27 2025 11:59PM. If you will need more time than this to complete your revisions, please reply to this message or contact the journal office at plosone@plos.org . Please include the following items when submitting your revised manuscript:

We look forward to receiving your revised manuscript.

Kind regards,

Khin Thet Wai, MBBS, MPH, MA

Academic Editor

PLOS ONE

Journal Requirements:

Additional Editor Comments:

To revise in accordance with the comments

Reviewers' comments:

Reviewer's Responses to Questions

**Comments to the Author**

1. If the authors have adequately addressed your comments raised in a previous round of review and you feel that this manuscript is now acceptable for publication, you may indicate that here to bypass the “Comments to the Author” section, enter your conflict of interest statement in the “Confidential to Editor” section, and submit your "Accept" recommendation.

Reviewer #1: (No Response)

Reviewer #2: All comments have been addressed

2. Is the manuscript technically sound, and do the data support the conclusions?

Reviewer #1: Yes

Reviewer #2: Yes

3. Has the statistical analysis been performed appropriately and rigorously? 

Reviewer #1: Yes

Reviewer #2: Yes

4. Have the authors made all data underlying the findings in their manuscript fully available?

Reviewer #1: Yes

Reviewer #2: Yes

5. Is the manuscript presented in an intelligible fashion and written in standard English?

Reviewer #1: Yes

Reviewer #2: Yes

6. Review Comments to the Author

Reviewer #1: The revised article has addressed most of my concerns. There are just two minor concerns that remain:

The source of covariate information is still not available in the methods. Are they from self-reported questionnaires, clinical measurements during the study, medical records, registers, or somewhere else?

Statistical methods: Thank you for clarifying that the authors have used log-binomial regression instead of logistic regression. I would suggest that the authors do not mention logistic regression, but instead just mention log-binomial regression.

Reviewer #2: (No Response)

7. PLOS authors have the option to publish the peer review history of their article (what does this mean? ). If published, this will include your full peer review and any attached files.

**Do you want your identity to be public for this peer review?** For information about this choice, including consent withdrawal, please see our Privacy Policy .

Reviewer #1: No

Reviewer #2: No

---

## [Author Response · Author response to Decision Letter 2]

13 Jan 2025

Many thanks for your comments.

We have now addressed the minor comments from the reviewers by updating the manuscript from ‘logistic regression’ to ‘log-binomial regression’ throughout and in line 185, we now specify that the covariates were ‘recorded in self-reported questionnaires.’

---

## [Editor Report · Decision Letter 2]

15 Jan 2025

PONE-D-24-17447R2Clusters and associations of adverse neonatal events with adult risk of multimorbidity: A secondary analysis of birth cohort data.PLOS ONE

Dear Dr. Macmillan-John,

Thank you for submitting your manuscript to PLOS ONE. After careful consideration, we feel that it has merit but does not fully meet PLOS ONE’s publication criteria as it currently stands. Therefore, we invite you to submit a revised version of the manuscript that addresses the points raised during the review process.

Please include the following items when submitting your revised manuscript:</please>

We look forward to receiving your revised manuscript.

Kind regards,

Khin Thet Wai, MBBS, MPH, MA

Academic Editor

PLOS ONE

Journal Requirements:

Additional Editor Comments :

Please do minor revisions as required.
---

## [Author Response · Author response to Decision Letter 3]

16 Jan 2025

Many thanks for your comments.

We have reviewed the reference list to ensure it is complete and correct.

We have also addressed the minor comments from the reviewers by updating the manuscript from ‘logistic regression’ to ‘log-binomial regression’ throughout and in line 185, we now specify that the covariates were ‘recorded in self-reported questionnaires.’

---

## [Editor Report · Decision Letter 3]

29 Jan 2025

Clusters and associations of adverse neonatal events with adult risk of multimorbidity: A secondary analysis of birth cohort data.

PONE-D-24-17447R3

Dear Dr. Macmillan-John,

We’re pleased to inform you that your manuscript has been judged scientifically suitable for publication and will be formally accepted for publication once it meets all outstanding technical requirements.

Kind regards,

Khin Thet Wai, MBBS, MPH, MA

Academic Editor

PLOS ONE

Additional Editor Comments (optional):

All essential comments are adequately addressed.
---

## [Editor Report · Acceptance letter]

PONE-D-24-17447R3

PLOS ONE

Dear Dr. John,

I'm pleased to inform you that your manuscript has been deemed suitable for publication in PLOS ONE. Congratulations! Your manuscript is now being handed over to our production team.

Kind regards,

on behalf of

Dr. Khin Thet Wai

Academic Editor

PLOS ONE